# The Relationship between Family Functioning and Pathological Internet Use among Chinese Adolescents: The Mediating Role of Hope and the Moderating Role of Social Withdrawal

**DOI:** 10.3390/ijerph18147700

**Published:** 2021-07-20

**Authors:** Xing-Kai Li, Pei-Shan Zhan, Shu-Dan Chen, Jie Ren

**Affiliations:** 1School of Education, Guangzhou University, Guangzhou 510006, China; 2111908089@e.gzhu.edu.cn (X.-K.L.); zhanpeishan@163.com (P.-S.Z.); 2111908009@e.gzhu.edu.cn (S.-D.C.); 2Huan Shi Road Primary School, Guangzhou 510095, China

**Keywords:** adolescents, family functioning, pathological internet use, hope, social withdrawal

## Abstract

This study constructed a moderated mediation model based on problem behavior theory to explore the psychological mechanism of family functioning interaction with pathological internet use. We used the Adolescent Pathological Internet Use Scale, General Functioning Scale, Trait Hope Scale, and Social Withdrawal Scale to measure internet use in 1223 middle school students. The results showed that (1) pathological internet use was negatively correlated with family functioning and hope, and positively correlated with social withdrawal; family functioning was positively correlated with hope, and negatively correlated with social withdrawal; hope was negatively correlated with social withdrawal; (2) family functioning could not only directly predict pathological internet use, but also indirectly predict pathological internet use through hope; and (3) the mediating effect of family functioning on pathological internet use was moderated by social withdrawal, which was stronger for individuals with low social withdrawal but not significant for individuals with high social withdrawal. This study revealed the internal mechanism of the relation between family functioning and adolescents’ pathological internet use, which has theoretical significance for improving adolescents’ hope and reducing their pathological internet use.

## 1. Introduction

The China Internet Network Information Center released the “National Research Report on Internet Use of Minors,” which showed that Chinese minor internet users reached 175 million in 2019. Currently, the internet penetration rate of minors is 93.1% in China [1]. The online activities most frequently reported by adolescents are social networking, schoolwork, entertainment, internet gaming, and online shopping [2]. The internet has become an indispensable tool for adolescents. However, internet addiction is one of the most common mental health problems among Chinese adolescents, and over 17% of them are addicted to the internet [3]. Internet addiction has severe adverse impacts on their sleep quality, academic achievements, and interpersonal relationships [4]. Given the complex definition and clinical diagnosis of internet addiction, the present study used pathological internet use (PIU) to indicate the irrational or improper use of the internet and the resulting negative consequences [5].

PIU has been proposed to be a type of problem behavior in adolescents [6]. Jessor’s problem behavior theory (PBT) defines risk behavior as anything that can interfere with successful psychosocial development and problem behavior as risk behaviors that elicit either formal or informal social responses designed to control them [7]. PBT identifies three systems in the theoretical framework of adolescents’ problem behaviors: personality, perceived environment, and behavior systems. Adolescents’ problem behaviors are generated amid a dynamic and continuous interaction between personality characteristics and the environment. PBT is a social-psychological framework that helps explain the development and nature of problem behaviors [8]. Under the PBT framework, our study examined the relationship between family functioning and PIU and its mechanism (the mediating role of hope and moderating role of social withdrawal).

### 1.1. Family Functioning and Pathological Internet Use

According to PBT, in the environment system, familial and parental factors (e.g., parental behaviors and attitudes) are the key to understanding adolescents’ problem behavior [9]. Family is a critical environment for adolescents to realize their physical and mental health. McMaster’s family functioning theory posits that a positive family environment plays an essential role in the healthy development of family members’ physiological, psychological, and social functions [10].

Family dysfunction (e.g., poor problem-solving strategies and ineffective communication among family members) means that the family system does not facilitate appropriate functioning [11]. In addition, it is a predictor of adolescents’ digital media abuse [4]. Typically, parents’ emotional connections, psychological support, and behavioral modeling help adolescents and satisfy the latter’s basic psychological needs [12]. However, adolescents with poor family functioning may seek support and resources from the internet. If adolescents’ behaviors are not effectively guided by their parents, the risk of PIU may increase [13]. Some studies have confirmed the direct relation between family functioning and PIU, with better family functioning seem to help reduce PIU in adolescents [14,15].

The role of cultural contexts also merits consideration. Compared with Western culture, Chinese culture values the importance of family more. Chinese parents tend to control their children and are highly involved in their children’s lives [12]. Family functioning may play an essential role in the occurrence of PIU among Chinese adolescents. Therefore, it is necessary to further investigate the relationship between family functioning and PIU as well as its psychological mechanism.

### 1.2. Hope as Mediator

Positive psychology believes that hope is an individual’s constructive cognition of the future [16]. Hope is also regarded as a positive motivational state, which is based on pathways thinking (the specific strategy to achieve a goal) and agency thinking (the motivation to achieve a strategy) [17]. Khazaei, Khazaei and Ghanbari-H [18] found that positive psychological interventions can effectively alleviate PIU. Self-esteem [14], self-efficacy [19], and other individual cognitive factors closely related to hope [17] have also been confirmed to be associated with PIU. The personality system of the PBT includes values, expectations, beliefs, and attitudes, reflecting the social learning and developmental experience of an individual [7]. Hope may be a protective factor for the personality system to reduce PIU in adolescents. Therefore, we hypothesized that hope is negatively correlated with PIU.

Hope, as an essential psychological capital, is also influenced by family conditions [20]. Suitable parent–child attachment [21] and family support [22] help enhance family members’ hope. Good family functioning can also provide a family environment with a sense of security, encouragement, and support, which promote the cultivation of adolescents’ objective and enterprising sense of hope [23]. Negative events in family life, such as neglect, physical abuse, or loss of parents, may make adolescents lose hope in the future [16,24]. Based on the above analysis, we hypothesized that family functioning is positively correlated with their sense of hope. Hope may play a mediating role in the relation between family functioning and PIU in adolescents.

### 1.3. Social Withdrawal as Moderator

Social withdrawal is used to describe solitary individuals’ behavior that preferring to stay at home and making minimal effort to engage in social activities [25]. In China, social withdrawal is a comprehensive concept that includes the withdrawal and inhibition shown in external behavior and the lack of internal communication motivation, tendency of loneliness, and emotional experience of shyness [26]. The relation between social withdrawal and internet use has attracted the attention of researchers. Tateno and colleagues [27] found that social withdrawal groups prefer to spend a considerable amount of time alone at home and use the internet excessively. Apart from social activities through the internet, they have very few social activities in real life. Long-term social withdrawal can make people increasingly lonely. Internet use (e.g., online games and social networking) can help alleviate loneliness but will also lead to PIU [28]. We therefore hypothesized that social withdrawal may be a risk factor for PIU. When family functioning and hope are protective factors of PIU, we aimed to explore whether adolescents’ social withdrawal can moderate the relation between family functioning and PIU, and the relation between hope and PIU.

Li and Wong [29] reported that maladaptive parenting and family dysfunction are critical factors in the development of social withdrawal; adolescents may not be able to communicate with others and establish good relationships in such family environments. In turn, social withdrawal can also impair adolescents’ social functions in school and in the family [30]. In addition, positive psychological qualities, such as resilience [31] and self-efficacy [32], are related to social withdrawal. Adolescents with severe social withdrawal may not have these positive psychological qualities. Therefore, we hypothesized that adolescents’ social withdrawal moderates the relation between family functioning and hope. Better family functioning will contribute to hope only in adolescents with fewer social withdrawal behaviors; conversely, the role of family functioning on hope will be weakened.

In this study, we proposed a moderated mediation model to explore the mechanisms of family functioning interaction with PIU (Figure 1). We examined whether hope has a mediating role between family functioning and PIU and whether social withdrawal has a moderating role in the mediation model.

## 2. Materials and Methods

### 2.1. Participants

Using a convenient sampling method, we selected 1223 students from six middle schools in Guangdong Province to participate in our study from September to November 2020. Informed consent was obtained from schools, students and their parents, and this study was approved by the ethics committee of Guangzhou University.

We excluded incomplete questionnaires (*n* = 6) and the score of all the questionnaires beyond three standard deviations (*n* = 120). A total of 1097 valid questionnaires were finally recovered (aged 10.25–19.58 years, M = 15.02, SD = 1.59), with an effective rate of 89.70%. The sample was composed of 514 boys (46.9%) and 583 girls (53.1%); 614 were junior (56%), and 483 were senior high school students (44%).

### 2.2. Measures

#### 2.2.1. Pathological Internet Use

We used the Adolescent Pathological Internet Use Scale (APIUS) developed by Li and Yang [33] to evaluate adolescents’ PIU. Based on Young’s [34] and Davis’s research [35], the authors believe that researchers can make a more accurate judgment of PIU from the cognitive, emotional, and behavioral symptoms. There are six core dimensions of PIU: salience, tolerance, compulsive internet use, mood alteration, social comfort, negative outcomes, which could fully reflect all aspects of adolescents’ pathological use of internet. In this 38-item, six-factor tool, each item is rated on a five-point scale ranging from 1 (totally inconsistent) to 5 (totally consistent). A higher score indicated a higher PIU severity. APIUS was moderately correlated with Young’s IAT, Chen’s CIAS and had good convergent validity [33]. The scale has a good measurement index among Chinese adolescent subjects. We conducted confirmatory factor analysis and found that the six-factor model fit the data well (χ^2^/df = 3.80, CFI = 0.89, TLI = 0.88, RMSEA = 0.05, SRMR = 0.05). The internal consistency of the full scale was 0.93, and that of each factor was 0.74, 0.71, 0.89, 0.87, 0.85, and 0.76, respectively.

#### 2.2.2. Family Functioning

The General Functioning Scale, developed by Epstein, Baldwin and Bishop [10], is a short version of the Family Assessment Device that can evaluate the level of family functioning. The scale has 12 items, of which six are used to describe beneficial family functioning, and six items describe obstructive family functioning. Each item is rated on a four-point scale ranging from 1 (very similar to my family) to 4 (not at all like my family). The higher the score, the better the family functioning. In this study, confirmatory factor analysis showed that the single-factor model fit the data well (χ^2^/df = 5.13, CFI = 0.92, TLI = 0.90, RMSEA = 0.06, SRMR = 0.04). The internal consistency of the questionnaire was 0.82.

#### 2.2.3. Hope

The Trait Hope Scale, developed by Snyder and colleagues [36], can be used to measure an individual’s positive expectation of their future. This 12-item tool has two factors, namely, agency thinking and pathway thinking, and is rated using a four-point scale, in which 1 = completely incorrect and 4 = completely correct. Higher scores indicate a higher sense of hope. In this study, confirmatory factor analysis showed that the two-factor model fit the data well (χ^2^/df = 4.20, CFI = 0.97, TLI = 0.95, RMSEA = 0.06, SRMR = 0.03). The internal consistency of the full scale was 0.79, and that of each factor was 0.65 and 0.71, respectively.

#### 2.2.4. Social Withdrawal

We used the Social Withdrawal Scale developed by Tian [25] to measure the social withdrawal of adolescents. The 16 questions cover three factors: avoidance of unfamiliar environment, outliers, and avoidance of speaking in public. Each item is rated using a five-point scale: 1 = completely inconsistent, 5 = completely consistent. The higher the score, the more serious the social withdrawal. The confirmatory factor analysis showed that the three-factor model fit our data well (χ^2^/df = 6.53, CFI = 0.95, TLI = 0.94, RMSEA = 0.07, SRMR = 0.06). The internal consistency of the full scale was 0.93, and that of each factor was 0.86, 0.92, and 0.84, respectively.

### 2.3. Data Analysis

We used Epidata 3.1 to input and manage raw data, and IBM SPSS Statistics for Windows, Version 25.0 to generate descriptive and correlation analyses on the observed variables. In Mplus8.3, we implemented the bias-corrected bootstrap estimation procedure to sample repeatedly 5000 resamples to test the mediating effect of hope. We used latent moderated structural equations to test the moderating effect of social withdrawal. Considering the influence of sex, age, and socioeconomic status (family monthly income, father’s education level, and mother’s education level were extracted as factors), we included them as control variables.

## 3. Results

### 3.1. Preliminary Analyses

Li and Yang define the APIUS project with an average score ≥3.15 points as a “pathological Internet use group”. In addition, they represent a group with an average score ≥3 points and <3.15 points as an “Approximate PIU group”. Those with an average score <3 points the group is defined as a “normal group of internet users” [33]. The detection rate of PIU in this study is 7.8%, which is close to the previous results [33]. The demographic difference test results of each variable are shown in Table A1.

The correlation coefficients of each latent variable are listed in Table 1. The mean value, standard deviation, and Pearson correlation coefficient for each observation variable are given in Table 2. The results showed that PIU was negatively correlated with family functioning and hope, and positively correlated with social withdrawal. Family functioning was positively correlated with hope, and negatively correlated with social withdrawal. Hope was negatively correlated with social withdrawal. The correlation analysis results indicated that the relation between the variables was in line with the hypotheses and met the conditions of the moderated mediation model test.

### 3.2. Testing for the Measurement Model

Our measurement model included four latent variables (PIU, family functioning, social withdrawal, and hope), and 14 observation variables (family functioning, including three observation variables after parceling; the parceling method was the balance in the factor method) [37]. We conducted confirmatory factor analysis on the measurement model, which was shown to have a good fit (χ^2^/df = 4.25, CFI = 0.97, TLI = 0.96, RMSEA = 0.06, SRMR = 0.04). The standardized load of each observed variable on the corresponding factor was significant (each *p* < 0.001), indicating that the observed variables could represent the latent variables well.

### 3.3. Testing for the Mediation Model

We tested the mediating effect of hope using mediating analysis based on structural equation modeling [38]. We established a simple regression model of latent variables to test the direct predictive effect of family functioning on PIU. The results showed that the model fit well (χ^2^/df = 5.19, CFI = 0.96, TLI = 0.95, RMSEA = 0.06, SRMR = 0.05). After controlling for sex, age, and socioeconomic status, we found that family functioning negatively predicted PIU (*β* = −0.35, *p* < 0.001). We then added hope as the mediating variable, and the results showed that the model fit well (χ^2^/df = 4.45, CFI = 0.96, TLI = 0.95, RMSEA = 0.06, SRMR = 0.05). As shown in Figure 2, the path coefficient between family functioning and PIU declined but still was significant (*β* = −0.28, *p* < 0.001). Family functioning had a significant effect on hope (*β* = 0.40, *p* < 0.001), whereas hope had a significant effect on PIU (*β* = −0.18, *p* < 0.001). The mediating effect of hope was −0.09 (*p* < 0.001, 95% CI: −0.15, −0.05) and accounted for 20.13% of the total effect. Therefore, hope had a significant mediating effect on family functioning and PIU.

### 3.4. Testing for the Moderated Mediation Model

According to our hypothesis, we used latent moderated structural equations to test the moderating effect of social withdrawal [39]. The first step was to test the benchmark model (bm) without latent interaction moderation. The results showed that the model fit well (χ^2^/df = 3.99, CFI = 0.95, TLI = 0.94, RMSEA = 0.05, SRMR = 0.04, LogL bm = −17001.63, AIC = 34135.26). The second step was to add the latent moderators (family functioning × social withdrawal and hope × social withdrawal) to the benchmark model to establish a moderated mediation model. The results showed that LogL mm = −16991.99, AIC = 34121.98. According to the formula LR (3) = 19.28 (*p* < 0.005), AIC decreased by 13.28. The fit of the moderated mediation model was better than that of the benchmark model.

Finally, we tested the moderated mediation model containing latent moderators (Figure 3 shows the moderating effect of social withdrawal). The latent moderators (family function × social withdrawal) had a significant effect on hope (*β* = −0.20, *p* = 0.001, 95% CI: −0.32, −0.08), but had no significant effect on PIU (*β* = −0.07, *p* = 0.245, 95% CI: −0.18, 0.05). The latent moderator (hope × social withdrawal) had a significant predictive effect on PIU (*β* = 0.11, *p* = 0.014, 95% CI: 0.02, 0.19), indicating that social withdrawal could moderate the relation between hope and PIU.

To demonstrate the moderating effect more intuitively, we divided social withdrawal into high and low groups according to ±1 standard deviation, and then performed a simple slope test. We examined the effect of family functioning on hope at different levels of social withdrawal (Figure 4). The results showed that when the level of social withdrawal was low (M-1 SD), family functioning had a more significant correlation with hope (*β*simple = 0.57, *p* < 0.001, 95% CI: 0.40, 0.74). When the level of social withdrawal was high (M + 1 SD), family functioning had a weak correlation with hope (*β*simple = 0.17, *p* = 0.053, 95% CI: −0.00, 0.33). The simple slope test (Figure 5) for the effect of social withdrawal on hope and PIU showed that, when the level of social withdrawal was low, hope significantly negatively predicted PIU (*β*simple = −0.18, *p* = 0.005, 95% CI: −0.31, −0.05, *p* = 0.05). When the level of social withdrawal was high, hope had no significant predictive effect on PIU (*β*simple = 0.03, *p* = 0.642, 95% CI: −0.10, 0.16).

Finally, for individuals with low social withdrawal (1 SD below the mean), the indirect effect was −0.11 (*p* = 0.008, 95% CI: −0.16, −0.05), and the proportion of indirect effect in the total effect was 36.67%. For individuals with high social withdrawal (1 SD above the mean), the indirect effect was 0.005 (*p* = 0.659, 95% CI: −0.04, 0.01), and the proportion of indirect effect in the total effect was 1.52%.

## 4. Discussion

PIU has been a topic of high interest in psychological research on adolescents. The present study proposed a moderated mediation model based on PBT. We comprehensively examined the relationship between the perceived environment system (family functioning), personality system (hope), and behavior system (social withdrawal) with adolescents’ PIU. The results showed that family functioning could directly predict PIU and indirectly predict it through hope. This mediating effect was particularly significant in adolescents with low social withdrawal.

### 4.1. Influence of Family Functioning on Pathological Internet Use: Mediating Role of Hope

Previous studies have shown that family factors may be closely correlated with PIU in adolescents [13,14]. Adolescents with better family functioning are less likely to have PIU. Families with high dysfunction typically display a lower ability to communicate and solve problems [11], compelling adolescents to seek support in the internet and internet overuse [15]. Therefore, good family functioning could be a protective factor in adolescents’ perceived environmental system against PIU. Zhong and colleagues have confirmed that family-based interventions could help enhance family functioning and reduce PIU [40]. Improving adolescents’ family functioning may help reduce the occurrence of PIU.

More importantly, we also confirmed the mediating role of hope between family functioning and PIU. Li and colleagues showed a significant negative correlation between family functioning and hope [23]. Consistent with the previous findings, we also found that family functioning has a positive effect on hope. Good family functioning helps enhance adolescents’ hope, which has a broad and far-reaching impact on student’s mental health and academic achievement [17]. We also found a significant negative correlation between hope and PIU. Adolescents with PIU often grow bored and feel lost when cutting internet use [19]. Adolescents who lack hope may prefer to indulge in the internet. Combined with Snyder’s hope theory, we believe that hopeful adolescents have more pathways and agency thinking to achieve their goals. Therefore, they are less likely to escape the real world by using the internet. Based on the above results, we infer that hope can be used as a protective factor in the personality system against PIU. The hope intervention plan based on the hope theory has been effectively applied in schools [41,42]. Teachers can set up positive psychology courses to help adolescents increase their sense of hope.

### 4.2. Moderating Role of Social Withdrawal

Our study verified the moderating role of social withdrawal in the proposed model. Specifically, the negative correlation between PIU and hope was more significant in low social withdrawal. Whether social withdrawal is a mental illness or a cultural phenomenon remains controversial [43]. Nonetheless, severe social withdrawal has adverse effects. Based on Tian’s study [26], we regard social withdrawal as a cultural phenomenon. Combined with PBT, we hypothesized that social withdrawal constitutes risk for problem behaviors in adolescents as a type that shows high involvement in other problem behaviors and low involvement in routine behaviors (e.g., learning and extracurricular activities). Our results show that when adolescents have a high degree of social withdrawal, the negative correlation between hope and PIU is not significant. Thus, helping adolescents reduce social withdrawal levels, rather than only enhancing their hope, maybe more helpful to reduce their PIU.

We also found that individuals with low social withdrawal higher values in hope covaried with better family functioning. Hamasaki [44] believed that family factors (e.g., lack of communication between parents) are considered risk factors for social withdrawal. Adolescents with dysfunctional families may find it challenging to learn to communicate with others and establish proper relationships in their families. When they fail to communicate with others, they may choose to close themselves off to avoid frustration. In addition, adolescents with social withdrawal often have the psychological characteristics of excessive dependence and maladjustment [29], which impede the formation of agency and pathways thinking necessary for fostering their sense of hope. Therefore, adolescents with high social withdrawal report feeling hopeless; even high levels of family functioning do not covary with higher hope levels in students.

However, we did not find a moderating effect of social withdrawal between family functioning and PIU. Regardless of the level of social withdrawal, family functioning is negatively correlated with PIU. Social control theory posits that when adolescents have a close relationship with their parents, they feel obliged to make their parents happy in a non-deviant way [13]. That might be why adolescents with high social withdrawal still had lower PIU if they had good family functioning.

### 4.3. Limitations and Implications

This study has some limitations. First, the conclusion is helpful for schools to take measures to help adolescents reduce PIU. However, students with a manifest internet use dependency (such as defined in the DSM) need professional help by clinical psychologists or psychiatrists (often in stationary treatment). Second, this study used a cross-sectional design, which could not examine the influence of family functioning, hope, and social withdrawal on PIU in different development periods. A longitudinal design would yield more meaningful data for further studies. Finally, our study did not test for psychopathologic attributes or features: depression, anxiety, impulsivity, perceived social support. In view of often occurrence of these features in adolescents with PIU, it is necessary to further test their role in the future.

## 5. Conclusions

In conclusion, this study explored the relationship between family functioning, hope, social withdrawal and PIU in adolescents. Based on the research results, we believe that good family functioning and hope can be used as protective factors against PIU in adolescents. Lower social withdrawal will help both factors play a better protective role. Therefore, the focus should be given to adolescents who have social withdrawal problems. Parents and educators should help them further integrate in class, establish good interpersonal relationships, and encourage them to have a more optimistic, hopeful view of their own future. These measures may also have a positive impact on reducing PIU in adolescents.

## Figures and Tables

**Figure 1 ijerph-18-07700-f001:**
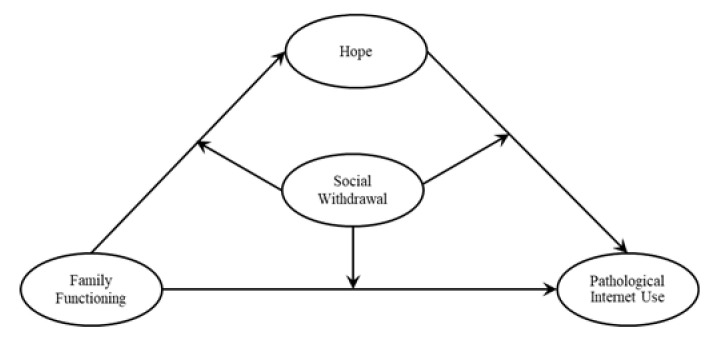
Research Hypothesis Model.

**Figure 2 ijerph-18-07700-f002:**
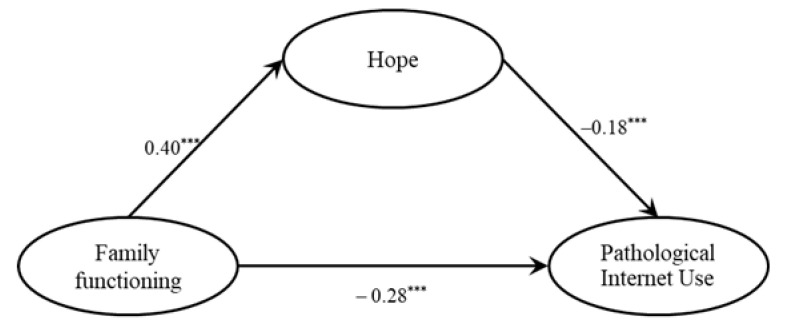
Test results of mediating effect of hope. Note: *** *p* < 0.001.

**Figure 3 ijerph-18-07700-f003:**
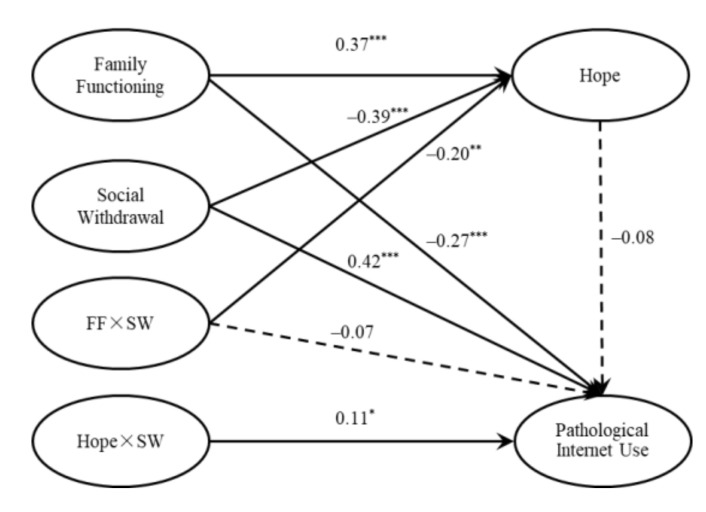
Results of moderating effect of social withdrawal. Note: **** p* < 0.001, ** *p* < 0.01, * *p* < 0.05.

**Figure 4 ijerph-18-07700-f004:**
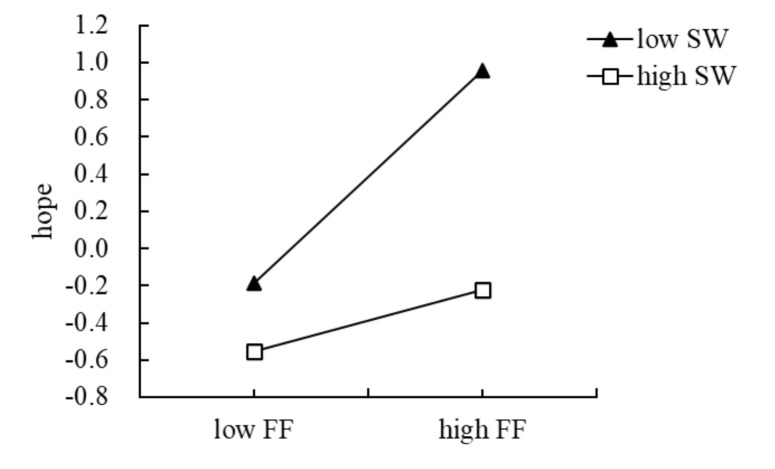
The relationship between family functioning and hope at two levels of social withdrawal: low SW = low social withdrawal, high SW = high social withdrawal.

**Figure 5 ijerph-18-07700-f005:**
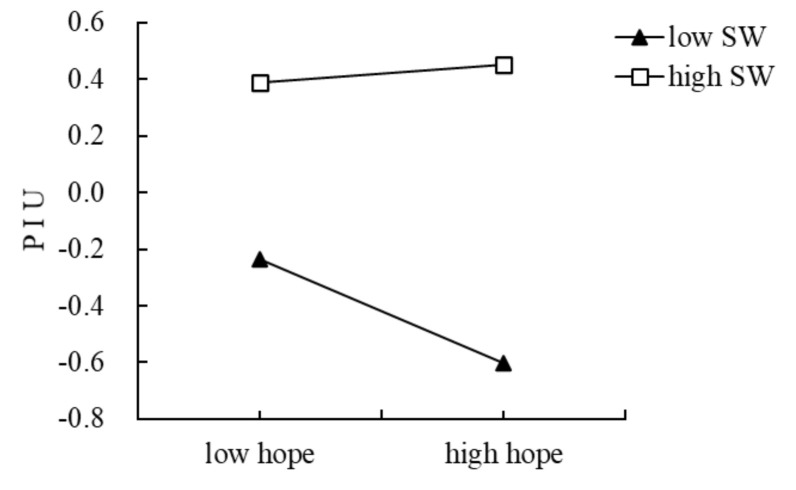
The relationship between hope and PIU at two levels of social withdrawal:low SW = low social withdrawal, high SW = high social withdrawal.

**Table 1 ijerph-18-07700-t001:** Correlation coefficients of latent variables.

Latent Variable	1	2	3	4
1. PIU	—			
2. Family functioning	–0.33 ***	—		
3. Hope	–0.32 ***	0.42 ***	—	
4. Social withdrawal	0.38 ***	–0.27 ***	–0.40 ***	—

Note: *** *p* < 0.001.

**Table 2 ijerph-18-07700-t002:** Correlation coefficients of observed variables.

	1	2	3	4	5	6	7	8	9	10	11	12	13	14	15	16	17
1. gender ^1^	—																
2. age	–0.03	—															
3. SES ^2^	0.09 **	–0.10 **	—														
4. PIU 1	0.12 ***	0.10 **	0.08 **	—													
5. PIU 2	–0.03	0.21 ***	0.04	0.49 ***	—												
6. PIU 3	0.04	0.16 ***	0.03	0.49 ***	0.66 ***	—											
7. PIU 4	0.04	0.08 *	0.07 *	0.35 ***	0.35 ***	0.49 ***	—										
8. PIU 5	0.04	–0.09 **	–0.02	0.25 ***	0.31 ***	0.42 ***	0.54 ***	—									
9. PIU 6	–0.02	0.34 ***	–0.03	0.46 ***	0.68 ***	0.58 ***	0.25 ***	0.24 ***	—								
10. FF 1	0.02	0.02	0.14 ***	–0.09 **	–0.18 ***	–0.24 ***	–0.13 ***	–0.26 ***	–0.26 ***	—							
11. FF 2	0.08 **	0.06	0.08 *	–0.06	–0.14 ***	–0.15 ***	–0.01	–0.11 ***	–0.20 ***	0.45 ***	—						
12. FF 3	0.04	–0.03	0.16 ***	–0.13 ***	–0.21 ***	–0.10 ***	–0.03	–0.13 ***	–0.29 ***	0.57 ***	0.50 ***	—					
13. Hope 1	0.19 ***	–0.09 **	0.11 ***	–0.03	–0.16 ***	–0.16 ***	0.02	–0.05	–0.16 ***	0.20 ***	0.22 ***	0.22 ***	—				
14. Hope 2	0.16 ***	–0.13 ***	0.12 ***	–0.12 ***	–0.25 ***	–0.25 ***	–0.09 **	–0.12 ***	–0.29 ***	0.27 ***	0.26 ***	0.24 ***	0.58 ***	—			
15. SW 1	–0.10 **	0.03	–0.08 **	0.12 ***	0.21 ***	0.23 ***	0.16 ***	0.16***	0.17 ***	–0.17 ***	–0.09 **	–0.11 ***	–0.25 ***	–0.26 ***	—		
16. SW 2	–0.04	0.14 ***	–0.04	0.20 ***	0.25 ***	0.21 ***	0.11 ***	0.16 ***	0.28 ***	–0.24 ***	–0.11 ***	–0.17 ***	–0.22 ***	–0.27 ***	0.57 ***	—	
17. SW 3	–0.04	0.07 *	–0.06	0.18 ***	0.27 ***	0.28 ***	0.17 ***	0.18 ***	0.28***	–0.22 ***	–0.12 ***	–0.17 ***	–0.27 ***	–0.33 ***	0.65 ***	0.534 ***	—
M	0.47	15.02	2.04	2.38	2.02	2.16	3.16	2.47	1.91	2.84	2.96	3.16	2.90	2.43	2.89	2.61	2.94
SD	–	1.59	0.70	0.87	0.76	0.76	0.96	0.87	0.66	0.57	0.51	0.54	0.50	0.55	0.92	1.01	0.91

^1^ Gender was a dummy variable, male=1, female=0, mean indicates the proportion of male. ^2^ SES: Socioeconomic status is formed by extracting a factor from the family monthly income, the education level of the father, and the education level of the mother. * *p* < 0.05, ** *p* < 0.01, *** *p* < 0.001.

## Data Availability

The raw data supporting the conclusions of this article will be made available by the authors, without undue reservation.

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
