# Peer review of "The Relationship between Family Functioning and Pathological Internet Use among Chinese Adolescents: The Mediating Role of Hope and the Moderating Role of Social Withdrawal"

_ijerph, 2021, doi:10.3390/ijerph18147700_

Round 1
Reviewer 1 Report
I think the topic is really interesting. There are some comments, however, I encourage the author to improve the article in many aspects. In the introductory section, in line 42 please describe in more detail the Theory of the Behavior Problem (PBT). The results section needs to be more detailed. For example, it is important to compare the studies in the literature with the results found from your own study. And what adds novelty to the scientific literature? Pico's question is unclear. Furthermore, I believe it is important to clarify how we could intervene to prevent these behaviors.
Author Response
Reviewer 1:
I think the topic is really interesting. There are some comments, however, I encourage the author to improve the article in many aspects. In the introductory section, in line 42 please describe in more detail the Theory of the Behavior Problem (PBT).
The results section needs to be more detailed. For example, it is important to compare the studies in the literature with the results found from your own study. And what adds novelty to the scientific literature? Pico's question is unclear.
Furthermore, I believe it is important to clarify how we could intervene to prevent these behaviors.
Thank you for your nice comments on our article.
- According to your suggestions, we give a more detailed explanation of PBT.
“Jessor’s problem behavior theory (PBT) defines risk behavior as anything that can interfere with successful psychosocial development and problem behavior as risk behaviors that elicit either formal or informal social responses designed to control them [7]. PBT identifies three systems in the theoretical framework of adolescents’ problem behaviors: personality, perceived environment, and behavior systems. Adolescents’ problem behaviors are generated amid a dynamic and continuous interaction between personality characteristics and the environment. PBT is a social-psychological framework that helps to explain the development and nature of problem behaviors [8]. Our study examined the relationship between family functioning (perceived environment) and PIU (problem behavior) and its mechanism (the mediating role of hope (personality characteristic) and moderating role of social withdrawal(personal behavior)) under the PBT framework”.
- According to your constructive suggestions, we have revised the manuscript extensively. At the end of the article, we explained the contribution of this study in detail, and supplemented the corresponding intervention studies.
Reviewer 2 Report
ijerph-1268229-peer-review-v1
The Relationship between Family Functioning and Pathological Internet Use among Chinese Adolescents: The Mediating Role of Hope and the Moderating of Social Withdrawal
Li, Zhan, Chen & Ren
Broad Comments
Moderator and mediator analyses are a field widely neglected by empirical research. This study analyzes the interdependencies of the Adolescent Pathological Internet Use Scale, the Family Functioning Scale, the Trait Hope Scale, and the Social Withdrawal Scale in a large adolescent sample. |
The study and its hypotheses are theory driven, which is rare enough. |
The study sample could be described in more detail, f.i. all scale means for boys/girls, for students younger than median age/older than median age, percentage of students that would be assessed with non-pathological scores and with pathological scores (PIU disorder). PIU cut-oof scores are of special interest. In case of space restrictions please add an appendix. These data are usually of special interest for epidemiologists, clinical psychologists, and psychiatrist. |
Authors tend to assume causal relationships between studied variables while there is covariation. In the Limitation section they hint to that, but in interpretation of their data they are not so clear. See below in Detailed feedback. |
They seem to propose that measures be taken that improve family functioning, social withdrawal, and hope (optimistic or positive future expectation). These measures are very vaguely described (“teachers should communicate”), the effectiveness of these measures [in the text as it is written now] is not disputed. Some may criticize that authors do not address questions of evaluation. |
Even if in China teachers are highly respected and appreciated as counsellors more than in European countries, should they not be given special training in family-based PIU counselling? And learn to assess when a student should be transferred to a clinical psychologist or a psychiatrist [best specialized in children and adolescents]? Social withdrawal is a very sensitive issue, because in many cases it will be confounded with psychopathological disorders; are teachers the right health care professionals here? |
Specific Comments
Line |
Text / remarks / suggestions / proposals |
11, 47 |
Mechanisms Þ more lenient would be “interplay” or “interaction” |
23 |
Revealed Þ “The study sheds light on the internal interplay…” Because the study provides a better and in-depth understanding on connections that are basically known, see Wansen, Li, Sui (2014). The relationship between recent stressful life events, personality traits, perceived family functioning and internet addiction among college students. 2014 Feb;30(1):3-11. doi: 10.1002/smi.2490. Epub 2013 Apr 25. |
42 |
Like in l.54 “McMaster’s family functioning…” Þ “Jessor’s problem behavior theory…” |
59 |
Predictor of adolescent’s addiction behaviors Þ too strong, the quoted article [4] only deals with mobile phone use, but not with substance abuse in general, not with PIU it would be more appropriate Þ “predictor of adolescent’s digital media abuse” |
64 |
The possibility of PIU Þ the risk of PIU, an elevated PIU risk |
91, 336 |
Ultimately Þ I would never miss this word on any part of this paper… |
94 |
Asocial Þ “solitary” would be more lenient and more common among European readers |
87, 114, 309,314 |
Meanwhile Þ I would delete it throughout the text, the use of this word (indicating a temporal gap = “in the meantime”) is very unusual |
100 158 |
Quotation: Tateno and colleagues Quotation: Snyder and colleagues |
114f. |
positive psychological qualities, such as resilience [30] and self-efficacy [31], are related to social withdrawal. Þ I would understand this easier in this rewording: “psychological qualities, such as resilience [30] and self-efficacy [31], are less related to social withdrawal” |
134f. |
Excluded … questionnaires beyond 3 SDs Þ 3 SDs in which scale? 3 SDs in the APIU? The Family Functioning Scale? … 3 SDs in any scale or in all scales? |
140ff. |
What is the theoretical/ clinical model for the APIU? APIU authors seem to refer to Davis: Davis RA (2001) A cognitive-behavioral model of pathological internet use. Comput Hum Behav 17: 187-195. doi:10.1016/S0747-5632(00)00041-8. European readers would think of a diagnostic tool based either on DSM or ICD criteria. Other approaches should be explained so that Europeans can understand it better. Maybe in an appendix it should be given what the APIU theoretical/ clinical model is, because reference [32; Li & Yang] seams not easily available for European readers. How is the APIU validated (construct validity, criterion validity)? Is it a screening tool with cut-off norms, or is it the operationalization of Davis’ model? A good understanding of the APIU is crucial for understanding the study results. |
151ff. |
Just a typo? It is not family function it is Þ family functioning Healthy/ unhealthy family function Þ “beneficial (or healthy) family functioning, obstructive (or unhealthy) family functioning” |
158 |
Trait Hope Scale … used to measure an individual’s hope level” is a bit circular Þ “… an individual’s positive expectation of his or her future” |
183 |
code. Þ just a typo? |
203 |
(ps <0.001) Þ I would have understood better: “(each p <0.001)” |
214 |
…declined but still significant… Þ “but still was significant” |
231 |
Finally, we tested the moderated mediation model containing latent moderators was tested Þ just a typo? was tested |
278f. |
When adolescents encounter problems, they should be able to seek more family support to reduce the adverse effects of PIU. This may be right but is no result of the study. Do authors want to propose that more family-based interventions should be implemented in PIU prevention and treatment? Then they should make it clear that they suggest this. |
284 |
We also found that hope can be used as a protective factor in the personality system against PIU. They could not have find this, because this is a correlational study in a cross-sectional design, no statements on “what can be used” or “what is effective” can be logically derived from this type of study. What authors probably want to say is that their findings encourage them for the assumption that hope may be a protective factor against PIU risks. |
286 |
Eliminating these barriers plays an essential role in reducing PIU. This no result of their study (see above). It can not be a result of a cross-sectional, questionnaire-based, non-interventional study to state what “plays an essential role” “in reducing PIU” (reduction is the effect of an intervention, either systematically in a professional treatment or naturalistic by environmental changes). The sentence would best be deleted. |
289ff. |
Teachers should communicate with parents to help students improve their family functioning and hope through positive psychological interventions [38], to reduce PIU. I understand that authors suggest to develop programs to improve teachers’ skills to help families improving their functioning, that would be the implementation (and evaluation) of a specific counseling training. This would help up to a point, but students with a manifest internet use dependency (like defined in the DSM) would need professional help by clinical psychologists or psychiatrists (often in stationary treatment). Authors should comment on this. |
300ff. |
Our results showed that even high hope does not help reduce PIU when a… See above: With their study design, their data/ results cannot “show” this. And that “high hope” reduces something is the causal assumption of “hope” being an agent that influences or impacts PIU, whereas the study analyses covariations only. |
303f. |
… found that individuals with low social withdrawal had a better effect of increasing their hope as family functioning improved… The meaning of this is unclear to me. Again, authors seem to assume causal effects. “…found that in individuals with low social withdrawal higher values in hope covaried with better family functioning…” Is this what authors mean? |
310 |
Therefore, adolescents with high social withdrawal show a lack of hope; even high levels of family functioning cannot affect their hope. Here is the same latent assumption of a causality: even in a well-functioning family a student might feel hopeless, maybe due to an undiagnosed psychological disorder. “Therefore, adolescents with high social withdrawal report feeling hopeless; even high levels of family functioning do not covary with higher hope levels in students.” |
320 |
The study did not test for psychopathologic attributes or features: depression, anxiety, impulsivity, perceived social support… This should be mentioned. |
335 |
...make them full of hope… “…encourage them so they have a more optimistic, hopeful view in their own future…” |
Author Response
Reviewer 2:
- Moderator and mediator analyses are a field widely neglected by empirical research. This study analyzes the interdependencies of the Adolescent Pathological Internet Use Scale, the Family Functioning Scale, the Trait Hope Scale, and the Social Withdrawal Scale in a large adolescent sample. The study and its hypotheses are theory driven, which is rare enough.
We feel great thanks for your professional review work on our article.
- The study sample could be described in more detail, f.i. all scale means for boys/girls, for students younger than median age/older than median age, percentage of students that would be assessed with non-pathological scores and with pathological scores (PIU disorder). PIU cut-oof scores are of special interest. In case of space restrictions please add an appendix. These data are usually of special interest for epidemiologists, clinical psychologists, and psychiatrist.
We feel great thanks for your professional review work on our article. According to your suggestions, we have added this information.
“Li and Yang define the APIUS project with an average score ≥ 3.15 points as a "pathological Internet use group", and define a group with an average score ≥ 3 points and < 3.15 points as an "Approximate PIU group", and those with an average score < 3 points the group is defined as a "normal group of internet users” [33]. The detection rate of PIU in this study is 7.8%, which is close to the previous results [33]. The demographic difference test results of each variable are shown in the Appendix 1.”
- Authors tend to assume causal relationships between studied variables while there is covariation. In the Limitation section they hint to that, but in interpretation of their data they are not so clear. See below in Detailed feedback.
We feel great thanks for your professional review work on our article. According to your suggestions, we have revised these contents.
- Even if in China teachers are highly respected and appreciated as counsellors more than in European countries, should they not be given special training in family-based PIU counselling? And learn to assess when a student should be transferred to a clinical psychologist or a psychiatrist [best specialized in children and adolescents]?
We feel great thanks for your professional review work on our article. As we know, in China, at least in cities and psychological teachers will receive professional psychological training to help adolescents solve psychological problems. For serious psychological problems, they will be evaluated in time and referred to psychiatrist for treatment.
- Social withdrawal is a very sensitive issue, because in many cases it will be confounded with psychopathological disorders; are teachers the right health care professionals here?
We feel great thanks for your professional review work on our article. Whether social withdrawal is a mental illness or a cultural phenomenon remains controversial [43]. In China, social withdrawal is a comprehensive concept that includes the withdrawal and inhibition shown in external behavior and the lack of internal communication motivation, tendency of loneliness, and emotional experience of shyness [26]. Based on Tian’s study [26], we regard social withdrawal as a cultural phenomenon. Therefore, we think teachers can help adolescents with social withdrawal tendency.
- Mechanisms Þ more lenient would be “interplay” or “interaction”
Thank you for your nice comments on our article. According to your suggestions, we changed it to "interaction".
- Revealed Þ “The study sheds light on the internal interplay…” Because the study provides a better and in-depth understanding on connections that are basically known, see Wansen, Li, Sui (2014). The relationship between recent stressful life events, personality traits, perceived family functioning and internet addiction among college students. 2014 Feb;30(1):3-11. doi: 10.1002/smi.2490. Epub 2013 Apr 25.
Thank you for your nice comments on our article. We read this study and revised our article.
- Like in l.54 “McMaster’s family functioning…” Þ “Jessor’s problem behavior theory…”
Thank you for your nice comments on our article. We changed it to "jessor's problem behavior theory...".
- Predictor of adolescent’s addiction behaviors Þ too strong, the quoted article [4] only deals with mobile phone use, but not with substance abuse in general, not with PIU it would be more appropriate Þ “predictor of adolescent’s digital media abuse”
Thank you for your nice comments on our article. According to your suggestions, we changed it to “predictor of adolescent’s digital media abuse”.
- The possibility of PIU Þ the risk of PIU, an elevated PIU risk
Thank you for your nice comments on our article. According to your suggestions, we changed it to “the risk of PIU, an elevated PIU risk”.
- Ultimately Þ I would never miss this word on any part of this paper…
Thank you for your nice comments on our article. According to your suggestions, We checked the full article and deleted “ultimately”.
- Asocial Þ “solitary” would be more lenient and more common among European readers
Thank you for your nice comments on our article. According to your suggestions, we changed it to “solitary”.
- Meanwhile Þ I would delete it throughout the text, the use of this word (indicating a temporal gap = “in the meantime”) is very unusual
Thank you for your nice comments on our article. According to your suggestions, We checked the full article and deleted “Meanwhile”.
- Quotation: Tateno and colleagues Quotation: Snyder and colleagues
Thank you for your nice comments on our article. According to your suggestions, We have revised these questions.
- positive psychological qualities, such as resilience [30] and self-efficacy [31], are related to social withdrawal. Þ I would understand this easier in this rewording: “psychological qualities, such as resilience [30] and self-efficacy [31], are less related to social withdrawal”
Thank you for your nice comments on our article. According to your suggestions, we changed it to “And psychological qualities, such as resilience [31] and self-efficacy [32], are less related to social withdrawal”.
- Excluded … questionnaires beyond 3 SDs Þ 3 SDs in which scale? 3 SDs in the APIU? The Family Functioning Scale? … 3 SDs in any scale or in all scales?
Thank you for your nice comments on our article. we changed it to “the score of all the questionnaires beyond 3 SDs”.
- What is the theoretical/ clinical model for the APIU? APIU authors seem to refer to Davis: Davis RA (2001) A cognitive-behavioral model of pathological internet use. Comput Hum Behav 17: 187-195. doi:10.1016/S0747-5632(00)00041-8.
European readers would think of a diagnostic tool based either on DSM or ICD criteria. Other approaches should be explained so that Europeans can understand it better. Maybe in an appendix it should be given what the APIU theoretical/ clinical model is, because reference [32; Li & Yang] seams not easily available for European readers. How is the APIU validated (construct validity, criterion validity)? Is it a screening tool with cut-off norms, or is it the operationalization of Davis’ model?
A good understanding of the APIU is crucial for understanding the study results.
Thank you for your nice comments on our article. According to your suggestion, we consulted the relevant literature and made the following revisions to the article. We used the Adolescent Pathological Internet Use scale developed by Li and Yang [33] to evaluate adolescents’ PIU. Based on Young’s [34] and Davis’s re-search [35], the author believes that researcher can make a more accurate judgment of PIU from the cognitive, emotional, and behavioral symptoms. Therefore, six core dimensions of PIU are proposed: salience, tolerance, compulsive internet use, mood alteration, social comfort, negative outcomes. In this 38-item, six-factor tool, each item is rated on a five-point scale ranging from 1 (totally inconsistent) to 5 (totally consistent). A higher score indicated a higher PIU severity. APIUS was moderately correlated with Young's IAT and Chen's CIAS, and had good convergent validity [33].
- Just a typo? It is not family function it is Þ family functioning Healthy/ unhealthy family function Þ “beneficial (or healthy) family functioning, obstructive (or unhealthy) family functioning”
Thank you for your nice comments on our article. According to your suggestions, we have corrected the errors.
- Trait Hope Scale … used to measure an individual’s hope level” is a bit circular Þ “… an individual’s positive expectation of his or her future”
Thank you for your nice comments on our article. According to your suggestions, we changed it to “… an individual’s positive expectation of his or her future”.
- Þ just a typo?
Thank you for your nice comments on our article. We deleted the error.
- (ps <0.001) Þ I would have understood better: “(each p <0.001)”
Thank you for your nice comments on our article. According to your suggestions, we changed it to “(each p <0.001)”.
- …declined but still significant… Þ “but still was significant”
Thank you for your nice comments on our article. According to your suggestions, we changed it to “but still was significant”.
- Finally, we tested the moderated mediation model containing latent moderators was tested Þ just a typo? was tested
Thank you for your nice comments on our article. We deleted the error.
- When adolescents encounter problems, they should be able to seek more family support to reduce the adverse effects of PIU.
This may be right but is no result of the study. Do authors want to propose that more family-based interventions should be implemented in PIU prevention and treatment? Then they should make it clear that they suggest this.
Thank you for your nice comments on our article. According to your suggestions, we changed it to “Zhong and colleagues have confirmed that family-based interventions have also been shown to enhance family functioning and reduce PIU [40]. Enhancing adolescents’ family functioning may help reduce the adverse effects of PIU”
- We also found that hope can be used as a protective factor in the personality system against PIU.
They could not have find this, because this is a correlational study in a cross-sectional design, no statements on “what can be used” or “what is effective” can be logically derived from this type of study. What authors probably want to say is that their findings encourage them for the assumption that hope may be a protective factor against PIU risks.
Thank you for your nice comments on our article. According to your suggestions, we changed it to “Based on above results, we infer that hope can be used as a protective factor in the personality system against PIU. The hope intervention plan based on the hope theory has been effectively applied in schools [41, 42]. Teachers can set up positive psychology courses to help adolescents increase their sense of hope”.
- Eliminating these barriers plays an essential role in reducing PIU.
This no result of their study (see above). It can not be a result of a cross-sectional, questionnaire-based, non-interventional study to state what “plays an essential role” “in reducing PIU” (reduction is the effect of an intervention, either systematically in a professional treatment or naturalistic by environmental changes). The sentence would best be deleted.
Thank you for your nice comments on our article. According to your suggestions, we deleted it.
- Teachers should communicate with parents to help students improve their family functioning and hope through positive psychological interventions [38], to reduce PIU.
I understand that authors suggest to develop programs to improve teachers’ skills to help families improving their functioning, that would be the implementation (and evaluation) of a specific counseling training. This would help up to a point, but students with a manifest internet use dependency (like defined in the DSM) would need professional help by clinical psychologists or psychiatrists (often in stationary treatment).
Authors should comment on this.
Thank you for your nice comments on our article. According to your suggestions, we add that: “This study has some limitations. First, the conclusion is helpful for schools take measures to help adolescents reduce PIU. However, students with a manifest internet use dependency (like defined in the DSM) would need professional help by clinical psychologists or psychiatrists (often in stationary treatment)”.
- Our results showed that even high hope does not help reduce PIU when a… See above: With their study design, their data/ results cannot “show” this. And that “high hope” reduces something is the causal assumption of “hope” being an agent that influences or impacts PIU, whereas the study analyses covariations only.
Thank you for your nice comments on our article. According to your suggestions, we changed it to “Our results show that when adolescents have a high degree of social withdrawal, the negative correlation between hope and PIU is not significant. Thus, helping adolescents to reduce social withdrawal level, rather than only enhancing their hope, may be more helpful to reduce their PIU”.
- … found that individuals with low social withdrawal had a better effect of increasing their hope as family functioning improved…
The meaning of this is unclear to me. Again, authors seem to assume causal effects.
“…found that in individuals with low social withdrawal higher values in hope covaried with better family functioning…” Is this what authors mean?
Thank you for your nice comments on our article. According to your suggestions, we changed it to “We also found that in individuals with low social withdrawal higher values in hope covaried with better family functioning”.
- Therefore, adolescents with high social withdrawal show a lack of hope; even high levels of family functioning cannot affect their hope.
Here is the same latent assumption of a causality: even in a well-functioning family a student might feel hopeless, maybe due to an undiagnosed psychological disorder.
“Therefore, adolescents with high social withdrawal report feeling hopeless; even high levels of family functioning do not covary with higher hope levels in students.”
Thank you for your nice comments on our article. According to your suggestions, we changed it to “Therefore, adolescents with high social withdrawal report feeling hopeless; even high levels of family functioning do not covary with higher hope levels in students“.
- The study did not test for psychopathologic attributes or features: depression, anxiety, impulsivity, perceived social support… This should be mentioned.
Thank you for your nice comments on our article. According to your suggestions, we add that: “Finally, our study did not test for psychopathologic attributes or features: de-pression, anxiety, impulsivity, perceived social support. In view of often occurrence of these features in adolescents with PIU, it is necessary to further test its role in the model in the future”.
- ...make them full of hope… “…encourage them so they have a more optimistic, hopeful view in their own future…”
Thank you for your nice comments on our article. According to your suggestions, we changed it to “…encourage them so they have a more optimistic, hopeful view in their own future…”

Reviewer 3 Report
The paper shows a bias in its conceptualisation, limiting itself exclusively to a context and bibliography. The authors do not describe in the theoretical framework the existing range of more recent internet addiction scales in the international context, such as those of Siomos 2009, Wölflin 2010, 2012, or other existing scales, such as those of Huang 2007, Chen 2003, Wang 2002, which are not cited in the paper.
Similarly, the discussion and conclusions are insufficient and are not addressed in the current literature.
Author Response
Reviewer 3:
The paper shows a bias in its conceptualization, limiting itself exclusively to a context and bibliography. The authors do not describe in the theoretical framework the existing range of more recent internet addiction scales in the international context, such as those of Siomos 2009, Wölflin 2010, 2012, or other existing scales, such as those of Huang 2007, Chen 2003, Wang 2002, which are not cited in the paper. Similarly, the discussion and conclusions are insufficient and are not addressed in the current literature.
We feel great thanks for your professional review work on our article. According to your suggestions, in the method part, we supplement the relationship between APIUS and those existing Internet use questionnaires.
“Based on Young’s [34] and Davis’s research [35], the author believes that researcher can make a more accurate judgment of PIU from the cognitive, emotional, and behavioral symptoms. There are six core dimensions of PIU: salience, tolerance, compulsive internet use, mood alteration, social comfort, negative outcomes, which could fully reflect all aspects of adolescents’ pathological use of internet.”
“APIUS was moderately correlated with Young's IAT and Chen's CIAS, and had good convergent validity [33]. The scale has a good measurement index among Chinese adolescent subjects.”
At the same time, we also modified the discussion to highlight the value of this study.
Thank you again for your positive comments and valuable suggestions to improve the quality of our manuscript.
Round 2
Reviewer 2 Report
Dear Sir or Madam,
the authors have reworked the text in a way that makes it ready for print: typos, notation, sample description, References section. The text is clearer and more concise, e.g. giving sufficient information of the central measure employed (APIU) and the theories the authors derive their hypotheses of. Tendencies to assume causal relationships between studied variables have been mitigated, and conclusions are more cautious.
Still some "cosmetic" details should be smoothed out in the paper. IMO there is no extra checking by referees necessary, I think the editorial staff will do this pretty well. If you decide otherwise, I will hear from you soon.
All other things being equal: stay healthy! Best regards, Peter-M.
Author Response
We feel great thanks for your professional review work on our article. According to the reviewer’s comments, we have revised the manuscript extensively. If there are any other modifications we could make, we would like very much to modify them and we really appreciate your help again.
Reviewer 3 Report
The authors have partially addressed the comments. It could be accepted at this stage
Author Response

(The authors gave the same response as above.)
